# Development of a clinical predictive score for allergic reactions during oral food challenges in pediatric patients

**Manatchanok Pattarakiatjaroen**[1], **Araya Yuenyongviwat**[2], **Pasuree Sangsupawanich**[2]*

1  Department of Pediatrics, Faculty of Medicine, Prince of Songkla University, Songkhla, Thailand,
2  Division of Allergy and Immunology, Department of Pediatrics, Faculty of Medicine, Prince of Songkla University, Songkhla, Thailand

* pasurees@gmail.com

## Abstract

### Background

The oral food challenge (OFC) is the gold standard for diagnosing food allergies; but, it carries risks ranging from mild to life-threatening reactions, including anaphylaxis. Understanding and predicting these reactions is critical for safe clinical practice.

### Objectives

This study aimed to develop and validate a clinically predictive score for allergic reactions during OFCs in pediatric patients.

### Methods

Using a retrospective database of pediatric patients undergoing OFCs at a pediatric outpatient clinic in Southern Thailand from January 2014 to December 2022, a multivariable predictive model was developed. Data on the reaction rates, demographics, and treatments were collected. Logistic regression analysis with predictor selection using a backward stepwise approach, was employed. The model's performance was assessed using the area under the receiver operating characteristic curve (ROC), calibration, and classification measures.

### Results

This study included 179 patients with an allergic reaction incidence of 12.3%. Predictors encompassed female, anaphylaxis and positive skin prick testing. The developed model achieved an ROC of 0.71. The patients were categorized into the low-risk (score 0–1) and high-risk (score 2–3) groups. Reaction rates during the OFCs were 6.7% in the low-risk group and 29.5% in the high-risk group.

**Data availability statement:** "All relevant data are within the paper and its Supporting information files."

**Funding:** The author(s) received no specific funding for this work.

**Competing interests:** None

## Conclusions

Our scoring model demonstrated predictive ability for OFC reactions in pediatric patients, offering valuable insights for clinical risk assessment.

## Introduction

The increasing prevalence of food allergies in children has led to significant physical and psychological challenges for both patients and their families. Managing food allergies often involves strictly avoiding certain foods and using alternative options, which can be difficult and expensive.

The progression of food allergies differs significantly, with over 50% of children typically outgrowing allergies to common foods like milk, eggs, wheat, and soy before reaching their teenage years [1]. Recognizing patients who have developed tolerance can reduce the impact of food allergies on individuals, families, healthcare systems, and society as a whole.

The oral food challenge (OFC) is considered the gold standard for diagnosing food allergies. During an OFC, a potential allergenic food is incrementally ingested under medical supervision to either confirm or exclude a food allergy diagnosis. However, OFC requires careful consideration because there is a risk of a systemic reaction [2].

In the past, it was recommended to use the 95% positive predictive value (PPV) level for specific IgE (sIgE) or skin prick tests (SPT) to assess the risk of reactions during OFCs [3]. However, this cannot be applied practically due to differences in cutoff values across various studies [4,5]. Prior investigation has combined OFC-related variables to create predictive models for systemic reactions. DunnGalvin et al. developed a logistic regression model using six variables: total and sIgE levels, SPT wheal size, age, sex, and symptom severity [6]. Although the predictive model achieved good performance, it required measuring total IgE level, which is not typically assessed as part of routine clinical practice in preparing for OFCs. Therefore, we aimed to develop a predictive model that utilizes routine clinical data available in real-world practice by incorporating various clinical parameters, such as patient history, demographic information, and standard test results.

## Materials and methods

### Participants

The medical records of pediatric patients under 18 years old who underwent OFCs at Songklanagarind Hospital between January 2014 and December 2022 were reviewed. The data were accessed on July 1, 2023. The study population included consecutive patients who underwent OFCs during the specified period.. Patient records with missing predictive data were excluded from the analysis. The Institutional Review Board of the Faculty of Medicine, Prince of Songkla University, Thailand, reviewed and approved this study (reference number 66-204-1-4). As this study is based on retrospective data; therefore, patient consent was not required by the Institutional Review Board to review their medical records. All data were maintained confidentially in accordance with the principles of the Helsinki Declaration.

### Predictors

The collected predictive data comprised four main components: 1) baseline demographics: sex and age; 2) personal allergic clinical presentations, including anaphylaxis, asthma, allergic rhinitis, atopic dermatitis, and drug allergies; 3) family history of allergy; and 4) laboratory

results within 1 month before OFCs: skin prick testing and serum-specific IgE levels related to the challenge food.

## Outcomes of OFCs

The open OFCs were performed in accordance with PRACTALL guidelines [7]. Allergic reactions during the OFC test were considered as outcomes. OFC reactions were documented and defined as any of the following systemic reactions with or without the medication needed: cutaneous, respiratory, cardiovascular, or gastrointestinal reactions. Cutaneous reactions included any type of rash or pruritus. Respiratory reactions included both upper and lower respiratory tract manifestations such as sneezing, rhinorrhea, nasal congestion, and wheezing. Cardiovascular symptoms included changes in heart rate, blood pressure, and skin color. Gastrointestinal symptoms included emesis, diarrhea, mouth itching, and abdominal discomfort.

## Sample size estimation

Following the methodology by Riley et al. [8] and assuming seven key predictors with an estimated concordance of 0.8, we calculated a minimum sample size of 173, including 17 events, based on a 10% event incidence from our pilot study. This target sample size aims to maintain a 0.05 difference between apparent and adjusted R-squared values, limits the intercept estimation margin of error to 0.05, and ensures sufficient events per predictor for model stability and accuracy.

## Statistical analysis

Statistical analyses were conducted using the Stata statistical software (version 18.0; StataCorp, Texas, USA). Descriptive statistics, such as frequency and percentage, were used for categorical data. Comparisons between two independent groups were performed using exact probability tests for categorical data. A two-tailed p-value <0.05 was considered statistically significant.

## Model development

Before developing the model, collinearity among predictor variables was assessed using variance inflation factors (VIF). A predictive model was created through multivariable stepwise backward logistic regression, incorporating predictors with a cutoff p-value of 0.1 from the univariable analysis while excluding those significant at 0.1. Weighted points were assigned according to their beta coefficients, calculated by dividing the logit coefficients by the smallest coefficient and rounding to the nearest integer. The model's discrimination performance was evaluated using the area under the receiver operating characteristic curve (ROC).

## Model validation

The internal validation of the predictive model was performed using bootstrapping techniques to randomize the dataset. This method enables repeated sampling from the data to generate multiple subsets, yielding a more accurate estimate of model performance. To evaluate the model's goodness-of-fit (GOF), the Hosmer–Lemeshow test was utilized. This test assesses how well the predicted probabilities align with the actual observed outcomes.

Additionally, a calibration plot was created to visually compare the model's estimated disease probabilities with actual observed occurrences. This plot helps assess the accuracy of predicted probabilities, demonstrating how well the model predicts disease likelihood across various risk levels.

## Results

During the research period, 214 patients underwent OFCs. However, 35 cases were excluded due to missing SPTs or specific IgE results, resulting in a sample of 179 patients for model development (Fig 1). Among these patients, 12.3% experienced allergic reactions during the OFCs. Four patients experienced severe reactions requiring treatment. There was a higher proportion of patients experiencing anaphylaxis, asthma and atopic dermatitis among those with failed OFCs. The prevalence of other allergic conditions, such as allergic rhinitis, atopic dermatitis, and drug allergies, was similar between individuals with passing and failing OFCs (Table 1). In addition to the clinical information, basic laboratory data, including SPTs and sIgE levels, were collected. It is worth mentioning that half of the children who experienced reactions to OFCs had positive SPT results. Table 1 outlines the ROCs, for the predictive factors identified in the study. Culprit foods of the study population undergoing OFCs are shown in S1 Table.

### Model development

Following a through analysis that included a variance inflation factor assessment to address multi-collinearity and stepwise logistic regression, it was found that none of the predictive factors needed to be removed because of multi-collinearity. Subsequently, the remaining predictive factors were gender, history of anaphylaxis, and positive skin prick testing. The ROC for the final model was 0.71, indicating that the model had discrimination ability.. The results, including the regression coefficient, odds ratio (OR), 95% confidence interval (CI), and item scoring for the clinical prediction of allergic reactions derived from the coefficients, are presented in Table 2. The scoring system ranges from 0 to a maximum of 3 points.

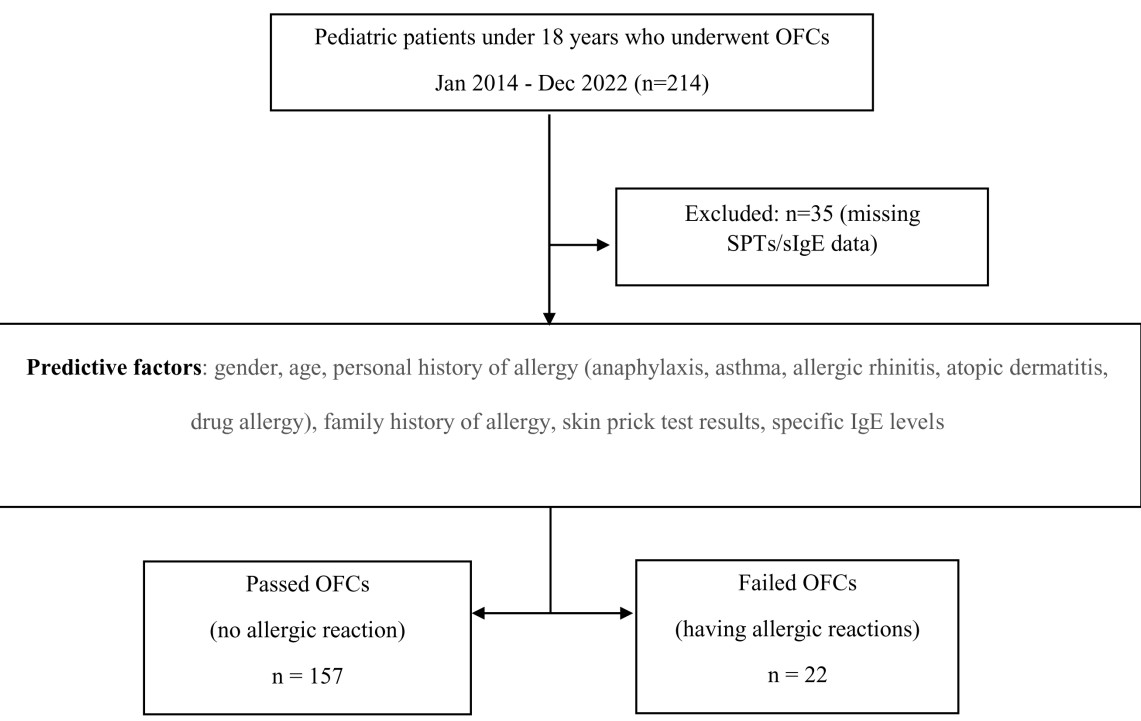

**Fig 1. Flow diagram of patient details.**

**Table 1. Baseline characteristics of patients performing oral food challenges.**

| Characteristics | No allergic reactions (N=157) | | Having allergic reactions (N =22) | | p-Value | ROC |
|---|---|---|---|---|---|---|
| | n | % | n | % | | |
| Age < 60 months | 115 | 73.2 | 16 | 72.7 | 0.959 | 0.50 |
| Female | 73 | 46.5 | 15 | 68.2 | 0.057 | 0.61 |
| Anaphylaxis | 31 | 19.7 | 9 | 40.9 | 0.026 | 0.61 |
| Asthma | 15 | 9.5 | 5 | 22.7 | 0.066 | 0.57 |
| Allergic rhinitis | 40 | 25.5 | 4 | 18.2 | 0.457 | 0.54 |
| Atopic dermatitis | 83 | 52.9 | 10 | 45.4 | 0.515 | 0.54 |
| Drug allergy | 14 | 8.9 | 3 | 13.6 | 0.480 | 0.52 |
| Family history of allergy | 12 | 7.6 | 3 | 13.6 | 0.342 | 0.53 |
| Positive skin prick testing | 40 | 25.5 | 11 | 50.0 | 0.017 | 0.62 |
| Specific IgE > 2 kUA/l | 20 | 12.7 | 6 | 27.3 | 0.070 | 0.57 |

**Table 2. Assigned score via multivariate logistic regression model for the prediction of allergic reactions after oral food challenges.**

| Parameters | Adjusted OR | 95% CI | p-Value | Coefficient | Assigned score |
|---|---|---|---|---|---|
| Female | 2.77 | 1.04-7.40 | 0.042 | 1.019 | 1 |
| Anaphylaxis | 2.39 | 0.88-6.51 | 0.089 | 0.871 | 1 |
| Positive skin prick testing | 2.52 | 0.96-6.62 | 0.061 | 0.924 | 1 |

## Score performance

The score performance was calculated via ROC, yielded a value of 0.71 (95% CI 0.56–0.83) (Fig 2). Additionally, the GOF was evaluated using the Hosmer–Lemeshow test, which indicated no evidence of lack of fit (p = 1.000). For internal validation, a bootstrapping method was employed, repeated 500 times, revealing a consistent ROC of 0.69 (95% CI 0.58, 0.80) with a bootstrap shrinkage of 0.92.

The risk curve illustrated in Fig 3 demonstrates that the predicted risk of allergic reactions increases proportionately with an increase in our proposed score. Additionally, the size of the circles within the curve represents the proportion of patients falling into each area. This visual representation suggests a clear association between the calculated score and the likelihood of allergic reactions, reinforcing the utility of our predictive model.

## Risk categories and score accuracy

The allergic reaction rate during OFCs for each score point is shown in S2 Table. Data show similar rate for score 0/1 and score 2/3. Therefore, patients were stratified into two groups based on their assigned scores: 0/1 for the low-risk group and 2/3 for the high-risk group. A higher score correlates with a greater prevalence of allergic reactions. Table 3 the allergic reaction rate was 6.7% in the low-risk group and 29.5% in the high-risk group.

With a cutoff predictive score of 2 or more, the scoring system has a positive likelihood ratio (LR+) of 5.87 (95% CI: 2.08–16.90, p < 0.01). This suggests that individuals scoring 2 or higher are nearly six times more likely to have an allergic reaction compared to the general population. This significant increase supports the score's utility in identifying high-risk individuals. On the other hand, the negative likelihood ratio (LR–) is 0.17 (95% CI: 0.06–0.48, p < 0.01). This reduction implies that individuals with a score below 2 have a substantially lower

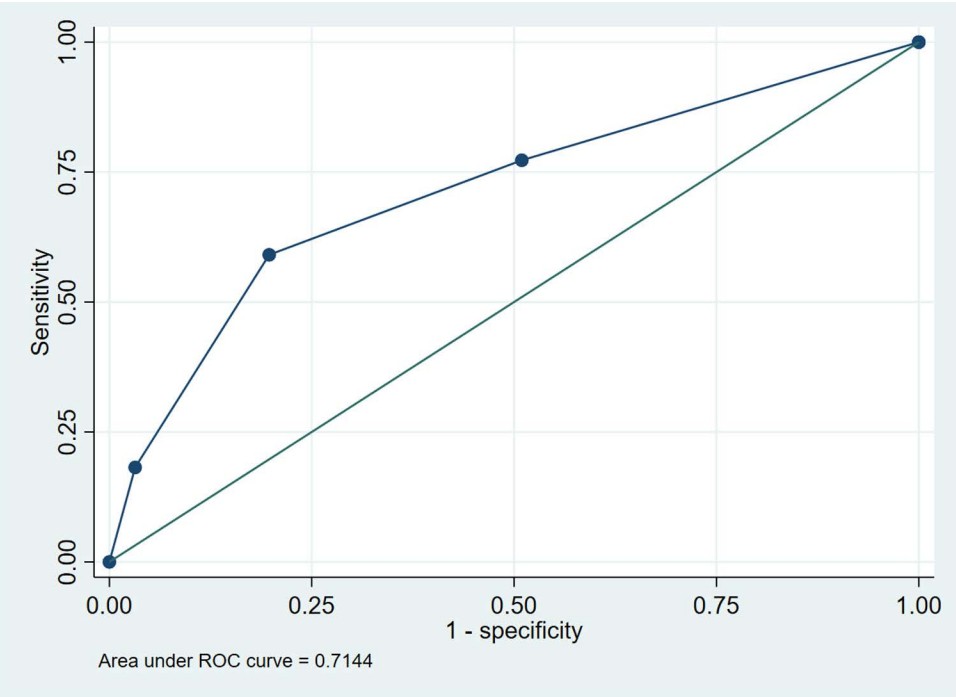

**Fig 2. The performance of the clinical predictive score evaluated using the area under the receiver operating characteristic (ROC) curve.**

likelihood of an allergic reaction compared to the general population, indicating the score's effectiveness in ruling out lower-risk cases.

## Discussion

Our final model utilized female gender, a history of anaphylaxis, and positive SPTs as key predictors for assessing the risk of allergic reactions during OFCs. The risk score derived from these predictive factors demonstrated good discrimination, as indicated by an ROC of approximately 0.71. By examining the allergic reaction rates according to each score point, patients were stratified into two groups based on their likelihood of allergic reactions: a low-risk group with a 6.7% probability and a high-risk group with a 29.5% probability.

DunnGalvin et al. [6] developed a logistic regression model aimed at predicting food allergy outcomes based on six key variables: total and sIgE levels, SPT wheal size, age, sex, and symptom severity. This model demonstrated strong predictive accuracy, achieving the ROC values of 0.97 for peanut, 0.95 for egg, and 0.94 for milk, underscoring its effectiveness. In comparison to our model, notable differences include the integration of total IgE as a predictive variable, along with the use of specific cutoffs for SPT and sIgE tailored to each type of allergenic food. A key challenge with employing these food-specific cutoffs, however, lies in the variability of cutoff values across different studies. This inconsistency can restrict the generalizability of the model's application across various populations and clinical settings.

In a prior study, a Food Challenge Score ranging from 0 to 4 was established, considering SPT wheal size, sIgE levels, history of non-cutaneous reactions, and age [9]. The notable distinctions between these and our scores were age and gender.Age did not significantly affect OFC outcomes in our data, similar to previous studies [10–12]. Research on the relationship

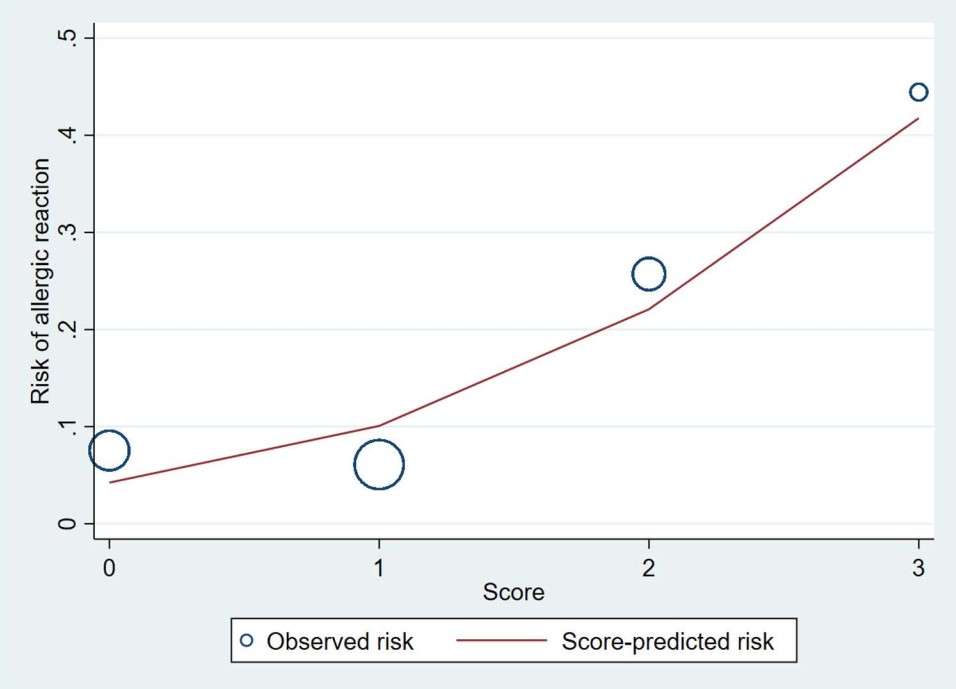

**Fig 3. The comparison between the predicted risk derived from the scoring system and the actual observed risk among patients.**

**Table 3. Distribution of allergic reactions in low and high probability categories.**

| Probability categories | Score | No allergic reactions | | Having allergic reactions | |
|---|---|---|---|---|---|
| | | n | % | n | % |
| Low | 0-1 | 126 | 93.3 | 9 | 6.7 |
| High | 2-3 | 31 | 70.4 | 13 | 29.5 |

between sex and outcomes of the OFC is limited. While no substantial evidence suggests a direct correlation between gender and the results of OFCs, some studies have noted differences in the prevalence and severity of food allergies between males and females [10]. However, these differences may not necessarily translate into variations in the OFC outcomes. More research is needed to understand the potential influence of gender on the outcomes of OFCs.

Based on our risk score, patients categorized in the low-risk group, with a reaction probability of 6.7%, may safely proceed with an oral food challenge (OFC) to obtain a definitive diagnosis, as their likelihood of experiencing a severe reaction is low. Conversely, for those in the high-risk group, with a reaction probability of 29.5%, the recommendation could be to delay the OFC or conduct it under close medical supervision, possibly requiring hospitalization to ensure immediate intervention if needed. This stratified approach not only aims to improve patient safety but also enhances the efficiency of managing suspected food allergies by tailoring the level of care to each patient's risk level.

Validation is crucial for the development of predictive models. While external validation, or evaluating the model's performance on separate data, is considered the gold standard

because of our limited study size, we internally validated our model using resampling procedures (bootstrapping), utilizing the same data for model development. The TRIPOD guidelines also caution against randomly splitting the data into two groups for model development and evaluation, stating that this approach is not recommended and may not be superior to the resampling approach [13].

## Conclusion

Our clinical predictive score offers promise for enhancing allergy services. Healthcare providers commonly evaluate the risk of OFC based on various factors, including an individual's medical history, current symptoms, and allergy test results. By incorporating the clinical predictive score into their assessment, healthcare providers can systematically assess and minimize the risk of allergic reactions during a challenge. This approach allows for a more standardized and evidence-based method for evaluating OFC risk, potentially improving patient outcomes and safety during food allergy testing.

## Supporting information

**S1 Table. Culprit foods of the study population undergoing OFCs.**
(DOCX)

**S2 Table. The allergic reaction rate during OFCs for each score.**
(DOCX)

## Author contributions

**Conceptualization:** Manatchanok Pattarakiatjaroen, Pasuree Sangsupawanich, Araya Yuenyongviwat.

**Data curation:** Manatchanok Pattarakiatjaroen.

**Methodology:** Manatchanok Pattarakiatjaroen, Pasuree Sangsupawanich.

**Writing – original draft:** Manatchanok Pattarakiatjaroen.

**Writing – review & editing:** Pasuree Sangsupawanich, Araya Yuenyongviwat.

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
