## [Decision Letter · Decision Letter 0]

30 Jul 2024

PONE-D-24-10955Development of a clinical predictive score for allergic reactions during oral food challenges in pediatric patients.PLOS ONE

Dear Dr. Sangsupawanich,

Thank you for submitting your manuscript to PLOS ONE. After careful consideration, we feel that it has merit but does not fully meet PLOS ONE’s publication criteria as it currently stands. Therefore, we invite you to submit a revised version of the manuscript that addresses the points raised during the review process.

The manuscript has been reviewed by our reviewer. Please find the comments of our reviewer and address carefully all the comments. ==============================

We look forward to receiving your revised manuscript.

Kind regards,

Firdaus Hamid

Academic Editor

PLOS ONE

https://linkinghub.elsevier.com/retrieve/pii/S2772829323001121

https://www.jaci-global.org/article/S2772-8293(23)00112-1/fulltext

https://journals.lww.com/co-allergy/abstract/2012/10000/guidelines_change_the_diagnostic_process_of_cow.20.aspx

In your revision ensure you cite all your sources (including your own works), and quote or rephrase any duplicated text outside the methods section. Further consideration is dependent on these concerns being addressed.

“None”

5. Please provide a complete Data Availability Statement in the submission form, ensuring you include all necessary access information or a reason for why you are unable to make your data freely accessible. If your research concerns only data provided within your submission, please write "All data are in the manuscript and/or supporting information files" as your Data Availability Statement.

Additional Editor Comments:

The manuscript has been reviewed by our reviewer. We suggest the author revise according to the reviewer's comments and suggestions.

Reviewers' comments:

Reviewer's Responses to Questions

**Comments to the Author**

1. Is the manuscript technically sound, and do the data support the conclusions?

Reviewer #1: Yes

Reviewer #2: Yes

2. Has the statistical analysis been performed appropriately and rigorously? 

Reviewer #1: Yes

Reviewer #2: Yes

3. Have the authors made all data underlying the findings in their manuscript fully available?

Reviewer #1: Yes

Reviewer #2: No

4. Is the manuscript presented in an intelligible fashion and written in standard English?

Reviewer #1: Yes

Reviewer #2: Yes

5. Review Comments to the Author

Reviewer #1: This interesting retrospective cohort study revealed the clinical predictive score for allergic reactions during oral food challenges in pediatric patients. This study concluded that the predictive score including female sex, anaphylaxis, positive skin prick testing, and specific IgE level > 2 kUA/l demonstrated a strong predictive ability for oral food challenge reactions in pediatric patients. For further clinical implications, here are my questions and suggestions.

Introduction

- Please overview the predictors for allergic reactions during oral food challenges identified in previous studies classified by food allergen and point out the knowledge gap leading to the hypothesis of this study.

Methods

- Please clearly describe the type of missing data leading to exclusion criteria.

- Were there any patients receiving oral immunotherapy before the oral food challenge?

- Did all patients perform skin prick tests, measurements of food-allergen-specific IgE and total IgE, and since when before the oral food challenge?

Results

- In demographics, please describe the types of food allergies and allergens to which the participants in this study were sensitized.

- Please describe the protocol for the oral food challenge you used in your study.

- How many patients had severe allergic reactions or food-induced anaphylaxis?

- Please describe how you select the cut-off predictive score and the Youden index.

- What is the AUC for the simple scoring model in comparison to the logistic regression model? Is it higher than a single predictor from positive skin prick testing or specific IgE?

- Apart from the prediction of positive oral food challenge, can it predict the threshold dose or severity of symptoms?

Discussion

- Please discuss the difference you found from previous studies developing predictive models for each food allergen such as egg, wheat, and milk allergy in pediatric patients.

- Please discuss the clinical implication of this predictive score

Reviewer #2: The author developed the clinically predictive score for allergic reactions during OFCs in pediatric patients in a single sit. Model development by multivariable logistic regression model and assessment of the model's performance was performed appropriately using standard statistical procedures. This work merit publication. However, there are some points to be added and mentioned.

1. Concerning the generalizability, types of allergic diseases among participants undertaken OFCs should be presented in the inclusion criteria and also in the result part (table1). Allergic reaction by OFCs should be detailed and summarized, according to the system involved.

2. According to risk categorization into low and high risk by the score, the author should suggest the clinical implication for further investigation or management in patients with different risk group. For example, the low risk group is not necessary to do OFC due to low probability?

3. The final logistic regression model in this study included both serum specific IgE and skin prick test. Were both test data available for all participants in this study? Is there collinearity problem? The final model demonstrated that specific IgE>2 is not statistically significant. Moreover, in clinical practice, it might be impractical to do both specific IgE and skin test to calculate the risk score. The author should explain why the final predictive model included these two test results.

6. PLOS authors have the option to publish the peer review history of their article (what does this mean? ). If published, this will include your full peer review and any attached files.

**Do you want your identity to be public for this peer review?** For information about this choice, including consent withdrawal, please see our Privacy Policy .

Reviewer #1: No

Reviewer #2: No

---

## [Author Response · Author response to Decision Letter 1]

7 Jan 2025

Reviewer #1: This interesting retrospective cohort study revealed the clinical predictive score for allergic reactions during oral food challenges in pediatric patients. This study concluded that the predictive score including female sex, anaphylaxis, positive skin prick testing, and specific IgE level > 2 kUA/l demonstrated a strong predictive ability for oral food challenge reactions in pediatric patients. For further clinical implications, here are my questions and suggestions.

Introduction

Please overview the predictors for allergic reactions during oral food challenges identified in previous studies classified by food allergen and point out the knowledge gap leading to the hypothesis of this study.

In the past, it was recommended to use the 95% positive predictive value (PPV) level for specific IgE (sIgE) or skin prick tests (SPT) to assess the risk of reactions during OFCs [3]. However, this cannot be applied practically due to differences in cutoff values across various studies [4,5]. Prior investigation has combined OFC-related variables to create predictive models for systemic reactions. DunnGalvin et al. developed a logistic regression model using six variables: total and sIgE levels, SPT wheal size, age, sex, and symptom severity [6]. Although the predictive model achieved good performance, it required measuring total IgE level, which is not typically assessed as part of routine clinical practice in preparing for OFCs. Therefore, we aimed to develop a predictive model that utilizes routine clinical data available in real-world practice by incorporating various clinical parameters, such as patient history, demographic information, and standard test results.

(Add this information to the last paragraph of the introduction.)

Methods

Please clearly describe the type of missing data leading to exclusion criteria.

Patient records with missing predictive data were excluded from the analysis.

(Add this information in the methods)

During the research period, a total of 214 patients underwent OFCs. However, 35 cases were excluded due to missing SPTs or specific IgE results, resulting in a sample of 179 patients for model development (Add this information in the results)

Were there any patients receiving oral immunotherapy before the oral food challenge?

None received oral immunotherapy prior to the oral food challenge.

Did all patients perform skin prick tests, measurements of food-allergen-specific IgE and total IgE, and since when before the oral food challenge?

All patients underwent skin prick tests (SPT) and specific IgE testing for suspected food allergens within one month before the oral food challenge. (Add this information in the methods)

Results

In demographics, please describe the types of food allergies and allergens to which the participants in this study were sensitized

The data is presented in Supplement 1.

Please describe the protocol for the oral food challenge you used in your study.

The open OFCs were performed in accordance with PRACTALL guidelines. (Add this information in the methods)

How many patients had severe allergic reactions or food-induced anaphylaxis?

Four patients experienced severe reactions requiring treatment.

(Add this information in the results)

Please describe how you select the cut-off predictive score and the Youden index.

The allergic reaction rate during OFCs for each score point is shown in Supplement 2. Data show similar rate for score 0/1 and score 2/3. Therefore, patients were stratified into two groups based on their assigned scores: 0/1 for the low-risk group and 2/3 for the high-risk group.

(Add this information in the results)

What is the AUC for the simple scoring model in comparison to the logistic regression model? Is it higher than a single predictor from positive skin prick testing or specific IgE?

AUC of a single predictor

• positive skin prick testing = 0.62

• specific IgE >2 kUA/l = 0.57

AUC of the final logistic regression = 0.71

AUC of the predictive score = 0.71

Apart from the prediction of positive oral food challenge, can it predict the threshold dose or severity of symptoms?

We were unable to develop a prediction model for severe reactions due to the small number of patients with severe reactions (n=4).

Discussion

Please discuss the difference you found from previous studies developing predictive models for each food allergen such as egg, wheat, and milk allergy in pediatric patients. compare previous study

DunnGalvin et al. [6] developed a logistic regression model aimed at predicting food allergy outcomes based on six key variables: total and specific IgE (sIgE) levels, skin prick test (SPT) wheal size, age, sex, and symptom severity. This model demonstrated strong predictive accuracy, achieving area under the curve (AUC) values of 0.97 for peanut, 0.95 for egg, and 0.94 for milk, underscoring its effectiveness. In comparison to our model, notable differences include the integration of total IgE as a predictive variable, along with the use of specific cutoffs for SPT and sIgE tailored to each type of allergenic food. A key challenge with employing these food-specific cutoffs, however, lies in the variability of cutoff values across different studies. This inconsistency can restrict the generalizability of the model’s application across various populations and clinical settings. (Add this information in discussion)

Please discuss the clinical implication of this predictive score implication

Based on the risk score, patients categorized in the low-risk group, with a reaction probability of 6.7%, may safely proceed with an OFC to obtain a definitive diagnosis, as their likelihood of experiencing a severe reaction is low. Conversely, for those in the high-risk group, with a reaction probability of 29.5%, the recommendation could be to delay the OFC or conduct it under close medical supervision, possibly requiring hospitalization to ensure immediate intervention if needed. This stratified approach not only aims to improve patient safety but also enhances the efficiency of managing suspected food allergies by tailoring the level of care to each patient’s risk level.

(Add this information in discussion)

Reviewer #2: The author developed the clinically predictive score for allergic reactions during OFCs in pediatric patients in a single sit. Model development by multivariable logistic regression model and assessment of the model's performance was performed appropriately using standard statistical procedures. This work merit publication. However, there are some points to be added and mentioned.

1. Concerning the generalizability, types of allergic diseases among participants undertaken OFCs should be presented in the inclusion criteria and also in the result part (table1). Allergic reaction by OFCs should be detailed and summarized, according to the system involved. suspected Food allergy ofc reaction.

The types of allergic diseases and allergic reaction rates is presented in Supplement 1.

2. According to risk categorization into low and high risk by the score, the author should suggest the clinical implication for further investigation or management in patients with different risk group. For example, the low risk group is not necessary to do OFC due to low probability? low do high postpone

Based on the risk score, patients categorized in the low-risk group, with a reaction probability of 6.7%, may safely proceed with an OFC to obtain a definitive diagnosis, as their likelihood of experiencing a severe reaction is low. Conversely, for those in the high-risk group, with a reaction probability of 29.5%, the recommendation could be to delay the OFC or conduct it under close medical supervision, possibly requiring hospitalization to ensure immediate intervention if needed. This stratified approach not only aims to improve patient safety but also enhances the efficiency of managing suspected food allergies by tailoring the level of care to each patient’s risk level.

(Add this information in discussion)

3. The final logistic regression model in this study included both serum specific IgE and skin prick test. Were both test data available for all participants in this study? Is there collinearity problem? The final model demonstrated that specific IgE>2 is not statistically significant. Moreover, in clinical practice, it might be impractical to do both specific IgE and skin test to calculate the risk score. The author should explain why the final predictive model included these two test results.

After comparing the model with and without sIgE, we observed no change in AUC. Given this lack of impact on predictive performance, we decided to exclude sIgE from the final model, and the results have been updated.

---

## [Editor Report · Decision Letter 1]

18 Mar 2025

Development of a clinical predictive score for allergic reactions during oral food challenges in pediatric patients.

PONE-D-24-10955R1

Dear Dr. Sangsupawanich,

We’re pleased to inform you that your manuscript has been judged scientifically suitable for publication and will be formally accepted for publication once it meets all outstanding technical requirements.

Kind regards,

Fatih Oz, Ph.D.

Academic Editor

PLOS ONE

Additional Editor Comments (optional):

Dear Authors,

Thank you for revising your article based on the reviewers’ suggestions. I am pleased to confirm that your article has been accepted for publication in Plos One. Congratulations.
---

## [Editor Report · Acceptance letter]

PONE-D-24-10955R1

PLOS ONE

Dear Dr. Sangsupawanich,

I'm pleased to inform you that your manuscript has been deemed suitable for publication in PLOS ONE. Congratulations! Your manuscript is now being handed over to our production team.

Kind regards,

on behalf of

Professor Fatih Oz

Academic Editor

PLOS ONE